# The Role of Metabolomics and Microbiology in Urinary Tract Infection

**DOI:** 10.3390/ijms25063134

**Published:** 2024-03-08

**Authors:** Haoyu Zheng, Chao Wang, Xiao Yu, Wenxue Zheng, Yiming An, Jiaqi Zhang, Yuhan Zhang, Guoqiang Wang, Mingran Qi, Hongqiang Lin, Fang Wang

**Affiliations:** 1Department of Pathogeny Biology, College of Basic Medical Sciences, Jilin University, Changchun 130021, China; zhenghy22@mails.jlu.edu.cn (H.Z.); wangc19@mails.jlu.edu.cn (C.W.); yuxiao23@mails.jlu.edu.cn (X.Y.); zhengwx22@mails.jlu.edu.cn (W.Z.); anym22@mails.jlu.edu.cn (Y.A.); zhangjiaqi0712@163.com (J.Z.); yhz23@mails.jlu.edu.cn (Y.Z.); wanggq20@jlu.edu.cn (G.W.); qimr@jlu.edu.cn (M.Q.); linhq@jlu.edu.cn (H.L.); 2Key Laboratory of Pathobiology, Ministry of Education, Jilin University, Changchun 130021, China

**Keywords:** urinary tract infection, metabolomics, microbiology, inflammation, pathogenesis

## Abstract

One of the common illnesses that affect women’s physical and mental health is urinary tract infection (UTI). The disappointing results of empirical anti-infective treatment and the lengthy time required for urine bacterial culture are two issues. Antibiotic misuse is common, especially in females who experience recurrent UTI (rUTI). This leads to a higher prevalence of antibiotic resistance in the microorganisms that cause the infection. Antibiotic therapy will face major challenges in the future, prompting clinicians to update their practices. New testing techniques are making the potential association between the urogenital microbiota and UTIs increasingly apparent. Monitoring changes in female urinary tract (UT) microbiota, as well as metabolites, may be useful in exploring newer preventive treatments for UTIs. This review focuses on advances in urogenital microbiology and organismal metabolites relevant to the identification and handling of UTIs in an attempt to provide novel methods for the identification and management of infections of the UT. Particular attention is paid to the microbiota and metabolites in the patient’s urine in relation to their role in supporting host health.

## 1. Introduction

The second-most frequent disease after respiratory disorders [1], urinary tract infection (UTI) is a widespread and very prevalent infectious disease globally [2]. UTI affects around 150 million people annually worldwide, according to data [3]. Annually, roughly 11 million cases are recorded in the United States alone, with direct and indirect expenditures totaling around USD 5 billion [1]. Because of their physiology and other factors, women are more likely than men to contract UTIs [4,5], with 12% of men and 50% of women reporting symptomatic UTIs at some point in their lives. Most UTIs occur in the bladder and are mostly caused by bacteria travelling up the urethra to the bladder. If cystitis is not properly treated, the bacteria can travel up to the kidneys and cause pyelonephritis, leading to scarring of the kidneys and causing kidney function impairment [6].

The current standard of treatment for UTI is to achieve urogenital sterility by eliminating the infection by antibiotic therapy [6]. The maintenance or restoration of the microbial communities that are present in a healthy condition of the host is not, however, a part of typical antibiotic techniques for the treatment of UTIs. This might be a result of the lack of understanding regarding the potential role that the urine microbiota plays in infection prevention. On the other hand, preserving host physiology and health is significantly aided by the microbial communities found in numerous bodily areas [7]. Bladder cancer, kidney stones, and other metabolic disorders are among the host health problems linked to the destruction of the human organism’s microbiota [8,9,10,11]. It is increasingly acknowledged that host and microbial populations are parts of a larger composite organism [12]. Because of this interdependence, modifications to the microbial ecology or host physiology may have an impact on the system as a whole; hence, modifications to the microbiota are essential for comprehending the emergence of UTIs.

Research on the pathogenesis of bacteria that has traditionally been conducted has focused on the pathogen’s particular virulence traits, such as toxins, adhesins, secretion and iron acquisition mechanisms, and defensive mechanisms against immune responses that are innate as well as adaptive. However, microbial metabolism is also a causative factor in the development of UTIs, and the ability of microbial metabolism is also directly related to which microorganisms are able to colonize a given environmental ecological niche. Metabolic interactions between microbes and hosts may play a role in UTI according to a growing corpus of studies [13,14,15]. Analyzing the metabolism of bacteria during infection is also essential for expanding our knowledge of pathogenesis and locating possible targets for cutting-edge antimicrobial medication development.

## 2. Human Microbiota and UTI

### 2.1. Normal Microbiological Composition of Urine

It used to be believed that the lower urinary system was sterile. The presence of bacteria is suggested by bacterial culture results from this environment is a sign that an infection is developing. However, the results of studies [16,17,18] have shown that bacteria and other microorganisms are also present in urine under healthy conditions. Traditional culture techniques favor fast-growing, mild bacteria, while fussy bacteria can avoid standard culture conditions and thus go undetected [19]. Using next-generation sequencing (NGS) for macrogenomic research makes it easier to quantitatively characterize the microbiome, providing insights into microbial populations and aiding in the identification of uncultured bacteria [20]. The use of enhanced quantitative urine culture (EQUC) techniques is advised in order to investigate the survivability of bacteria in urine, as the DNA sequencing-based approach has a disadvantage in that it is unable to demonstrate the vitality of detected bacteria [21]. As a result, the integration of EQUC and 16S rRNA gene sequencing will yield a trustworthy characterization of the urinary tract (UT) microbial population, as culture-based techniques will enable an approximative measurement of bacteria and, hence, function as indicators with practical significance.

With the advancement of our knowledge of the human microbiome, resident microbial communities of the UT of humans have been identified. The human upper tract has a low biomass in comparison with numerous different microbial ecological niches [22]. Price et al. [23] demonstrated through a cross-sectional study that the most common microbiota was *Lactobacillu*. Moreover, a decline in the quantity of *Lactobacillus* is associated with a pathological state; *Lactobacillus* maintain an acidic environment, so its impairment favors the colonization of pathogenic uropathogens [18,24]. *Gardnerella* follows closely, in terms of abundance [25], and various urogenital urobacterial communities have been identified, named after their respective dominant genera, such as *Lactobacillus*, *Streptococcus*, *Bifidobacterium*, *Staphylococcus*, etc. (Figure 1 (right), urethral microbiota of healthy female). Certain microbiota types, such those dominated by *Gardnerella vaginalis*, *Atopobieum vaginae*, and *Lactobacillus crispatus*, are exclusive to healthy women, whereas others are commonly linked to certain disorders. In contrast, the *Gardnerella* and *Escherichia* microbiota are more common in younger and older women. Female age, as well as hormone levels, also have an impact on the diversity of urogenital microorganisms [23]. Additionally, it was shown that in the elderly, the amount of *Bifidobacteria* decreased with age, the relative abundance of *Lactobacilli* decreased, and the relative abundance of *Peptococcus* increased [26]. In contrast, genera such as *Mobiluncus*, *Oligella* or *Porphyromonas* are common in postmenopausal women [27].

### 2.2. Microbiota Structure of UTI-Infected Patients

#### 2.2.1. Changes in the Urinary Microbiome and Pathogenesis in UTIs

UTI formation and the makeup of the urine microbiota in people are starting to be intensively researched [28]. Traditionally, normal urine culture procedures used in clinical microbiology labs have been the basis for the identification of UT bacteria. Merely a small quantity of microbes—mostly aerobic, quickly growing bacteria like *Escherichia coli* (*E. coli*)—can be detected using these techniques [20]. But *E. coli* is not the only type of bacteria that causes UTIs; this approach also misses anaerobic microorganisms with sluggish growth or those with complex nutritional needs, leading to false-negative diagnoses. Currently, with the development of histological techniques, more and more researchers are analyzing changes in the microbiome of people with UTIs to better reveal the mechanisms of disease development. The quantitative nature of the microbial whole genome sequencing (WGS) method identifies traditional uropathogens in low numbers in samples characterized by infection. Gilbert et al. [29] used mouse bladder RNA-Seq data to point out the shedding and differentiation of uroepithelial cells affected by *Gardnerella* exposure, promoting the persistence of uropathogenic *E. coli* (UPEC) in the bladder. Nonetheless, research is ongoing to fully understand the precise mechanisms driving the development and maintenance of the urinary microbiota. The development of reliable protocols to be used in clinical and research facilities is the main focus of current advancements in this field [21]. Moustafa et al. [30] successfully analyzed 116 samples by 16S rDNA sequencing and found that abnormal proliferation of microbiota such as *Aspergillus* and *E. coli* was found in the patients. An increasing number of researchers have confirmed changes in the microbiome in UTIs, including abnormal proliferation of bacterial communities and reduced microbial diversity [22,31,32] (Figure 1 (right), urethral microbiota of female UTI patients). According to this research, there are a number of contributing variables to UTIs. The UT biome’s makeup seems to be more important for UT defense, and an imbalance in it might be just as likely to cause illness as the UT pathogens that are invading the area.

#### 2.2.2. Gut Microbiota and UTI

The current study demonstrates a novel connection between the bacteriuria’s development and the gut microbiome and UTIs in the future, as well as supporting the theory that UT pathogenic bacteria start in the stomach. Because resident bacteria have been proven to impact the critical activities of several human organs, the human microbiome is thought to play a significant role in the start and progression of a wide range of disorders [33]. The current theory on the genesis of the female urine microbiota suggests that it might come from the environment, the vaginal microbiota, and the gut microbiota. Research indicates that there may be a possibility of bacterial migration from the stomach to the UT due to the correlation between the gut microbiota and the urinary microbiota’s diversity and composition [28].

Uropathogenic bacteria such as UPEC are thought to have their primary host in the gut, which also serves as a staging region for the bacterium’s entry into the mucosal ecotone of the proximal genitourinary tract, where it may then climb to the bladder and cause UTIs [34,35] despite the fact that uropathogenic bacteria can become long-term hosts inside the epithelium of the human bladder [36]. Magruder et al. [37] conducted serial gut microbiota investigations in a pilot study including 26 recipients of renal transplants and found a correlation between gut *Enterococci* abundance and UTI. Additionally, a number of studies have demonstrated that the gut microbiome affects the development of UTIs, with increased abundance of intestinal *E. coli* or *Enterococci* in patients with UTIs compared to those without UTIs [38,39]. Furthermore, Worby et al. [40] discovered that, in comparison to healthy controls, women with recurrent UTI (rUTI) had intestinal microecological dysbiosis, which was defined by decreased microbial abundance and butyric acid-producing bacterial levels.

#### 2.2.3. Vaginal Microbiota and UTI

Because specific bacteria present in the vagina may also be found in the urine, women’s physiologies may contribute to the vaginal microbiota’s influence over the urinary microbiota [41]. The host’s susceptibility to UTI may be influenced by vaginal flora. If probiotics, especially *Lactobacillus crispatus*, change the vaginal microbiome, women with rUTI become more resistant [42]. Furthermore, women with a healthy microbiota mostly composed of lactic acid bacteria are less likely to develop a UTI than women with bacterial vaginosis caused by anaerobic overgrowth, such as some strains of *Gardnerella* vaginalis [43]. It has been demonstrated that brief exposure to specific strains of *Gardnerella* vaginalis activates *E. coli* in the bladder’s dormant intracellular reservoirs, increasing the risk of rUTI by inducing interleukin 1 receptor-mediated damage and apoptosis in bladder epithelial cells [44]. When combined, these findings broaden our understanding of the pathophysiology of UTIs and raise the possibility that the illness is brought on by sporadic contact with intestinal or vaginal-associated bacteria, which are not usually thought to be uropathogenic.

#### 2.2.4. Environmentally Altered Microbiota and UTI

A non-negligible contributing factor to UTIs, catheter implantation causes a range of immunological and histological changes in the bladder; catheterization modifies the ecology of the bladder, causing inflammation and fibrinogen production, and fosters an environment that is more conducive to the colonization of common UTI pathogens. *Enterococcus* through endocarditis- and biofilm-associated pilus (Ebp) hyphae with a fibrinogen-binding adhesin called EbpA at the tip binds to fibrinogen encapsulating the conduit and overcomes neutrophil-mediated immune responses [45]. *Enterococcus* also uses fibrinogen as a source of nutrition, enabling it to grow in urine [46]. The virulence of *Enterococcus* is dependent on two secreted proteases, gelatinase (GelE) and a serine protease (SprE), which are upregulated during infection to cleave the α, β, and γ chains of fibrinogen, thereby enhancing biofilm formation [47]. Less than 3% of all positive urine cultures had *Staphylococcus aureus*; yet, this kind of urine is more likely to cause bacteremia. [32,48]. The most significant risk factor for complex methicillin-resistant *Staphylococcus aureus* (MRSA) UTI is urinary catheterization [32]. There is evidence that clumping factor B (ClfB) actively participates in the pathophysiology of UTIs [49]. Meanwhile, *Staphylococcus aureus* promotes fibrinogen release and bacterial persistence in the bladder by upregulating the expression of IL-1α, IL-1β, IL-6, and TNF, which enhances catheter-mediated inflammation [49]. Therefore, it is an emerging cause of complex UTIs, including infected kidney stones [50]. *Proteus mirabilis* is a common component of the UT microbiota in patients with urinary catheters [51]. *Proteus mirabilis* adheres to the bladder using flagella and adhesive mannose-resistant *Proteus*-like (MR/P) fimbriae [52]. In the meantime, *Proteus mirabilis* produces large amounts of urease, an enzyme that may increase urine’s pH and hydrolyze urea to ammonia. This process allows for the creation of crystalline biofilms, which both maintain an ecological niche during antibiotic therapy and encourage the return of sickness [53] (as shown in Table 1, pathogenesis of catheter-associated UTIs).

### 2.3. Relationship between Microbiome and UTI

UTIs are among the most typical illnesses that afflict the physical and mental health of patients in the emergency setting and are associated with problems such as the long time required for urine bacterial cultures, the ineffectiveness of empirical anti-infective treatments in the emergency setting, and the tendency of the disease to deteriorate into rUTI. At least as many microbes as somatic cells colonize the human body [54], and the use of next-generation sequencing technology has made it possible to characterize the urine microbiome and disprove the conventional wisdom regarding a sterile bladder [55].

The bacteria in the UT [53,56], the digestive tract [40], and the vagina [57] all have an impact on the development, duration, and recurrence of UTIs [58,59]. Urobiomes, which are often dominated by a single species or genus, have been seen in a variety of urotypes. Urogenital types are classified according to their dominating taxon, which is *Lactobacilli*. Other urogenital types include those with no prominent taxon or those dominated by bacteria such as *Gardnerella*, *Prevotella*, and *Streptococcus.* Numerous subtypes of the genitourinary microbiome have been identified, and they resemble the urinary system’s composition and distribution. Furthermore, it is important to investigate the roles played by the fungal [60] and viral [61] groups in the UT environment, since they can potentially affect the microbiome. Interactions between other microbial populations, including those in the stomach and vagina, can potentially lead to UTI development [62]. Similar microbiota are revealed by macrogenomic sequencing of the female UT and vagina, indicating cross-community interactions [63]. The idea of a gut microbiota–UTI axis has been supported by recent investigations of human feces, which have also revealed a strong relationship between *E. coli* strains from the gut and urine [37].

The positive function that microbes play in preserving homeostasis in the human body is acknowledged. Preventing infections and preserving a normal pH in the vagina is largely dependent on genitourinary microbiota. By creating a physical barrier against pathogens, microbiota play a critical defensive role and aid in the immune system’s development [64]. Antibiotic usage can be erratic, particularly in female patients with rUTI. This can lead to a rise in antibiotic resistance in the pathogenic organisms [65]. Therefore, research into the urinary microbiome of healthy people and how it contributes to disease by allowing opportunistic pathogens to proliferate could lead to new discoveries regarding the diagnosis and treatment of urological pathology.

## 3. Urinary Metabolome and UTI

### 3.1. Common Methods and Techniques for Studying Metabolites in Urine

In most clinical laboratories, urine culture is currently the procedure of choice for diagnosing UTIs. However, this method has a relatively long culture cycle, the time taken to obtain culture results is usually longer than that required to initiate antimicrobial therapy [66], and there is no simplified method of classifying the etiology of bacterial UTI [67]. Most human diseases are characterized by changes in humoral metabolites before and during the onset of clinical symptoms [68]. Metabolomics is used to predict the levels of metabolites in the body and to diagnose the physiological state of the organism in time to guide appropriate stimulation or pharmacological intervention [69]. With great accuracy and repeatability, metabolomics may be used to diagnose the bacterial cause of UTIs [70]. The body’s metabolic breakdown products are present in urine, which is easily obtained in large amounts and might represent an organism’s status [71]. Urinary metabolomics is now transitioning from the discovery phase to the validation phase by using urine composition to diagnose disorders [72].

Nowadays, most studies employ thorough analysis based on one-dimensional separation-liquid chromatography–tandem mass spectrometry (1D-LC-MS/MS) and two-dimensional separation-liquid chromatography–tandem mass spectrometry (2D-LC-MS/MS) [73], high-resolution nuclear magnetic resonance (NMR) spectroscopy [74], high-performance liquid chromatography/time-of-flight mass spectrometry (HPLC-TOFMS) [75], liquid chromatography–quadrupole exactive high-field orbitrap mass spectrometry (LC-Q Exactive HF MS) [76], high-performance liquid chromatography (HPLC) [41], high-performance liquid chromatography–tandem mass spectrometry (HPLC-MS/MS) [77], ultra-high-performance liquid chromatography-high-resolution orbitrap mass spectrometry (UHPLC-HRMS) [78], nanoflow liquid chromatography-electrospray ionization–tandem mass spectrometry (LC-ESI-MS/MS) [79] and liquid chromatography coupled with electrospray quadruple time-of-flight mass spectrometry (LC/QTOF-MS) [80], and other means to experiment on human urine samples. Untargeted metabolomics is mostly dominated by the application of a single analytical platform, and each of these technologies has unique strengths and limitations. Overall, NMR and MS detect metabolites of different chemical classes with minimal overlap and are, therefore, highly complementary in metabolomics applications. This multiplatform metabolomics study enables us to discover and quantify a wide range of distinct urine metabolites or metabolite species, boosting metabolome coverage. Numerous research groups also combine two or more analytical approaches to assist enhance metabolite coverage [74,81].

### 3.2. Effect of UTIs on the Metabolome

The most abundant catabolic metabolites in normal human urine are usually urea, citrate, amino acids, and small peptides [82]. Since *E. coli*-associated UTIs account for 75–90% of all infections, they are the most prevalent kind of UTIs overall [83]. For this reason, studying its metabolites is important for both UTI diagnosis and therapy. Infected urine samples have always been shown to include bacterial metabolic end products [84]. A group of these metabolites have been investigated and suggested as indicators for bacterial UTIs. Most studies [70,85] have found that urine acetic acid and malodorous amine, trimethylamine (TMA) are the best urine biomarkers for bacterial UTIs and *E. coli*-associated UTIs. This is because the metabolic consequences of bacterial urine contamination in the setting of chronic illness might be linked to their existence. It is known that trimethylamine N-oxide (TMAO) is reduced by microorganisms using bacterial TMAO- reductase in the bladder, which results in the production of TMA [70,86,87]. This makes TMA a unique indicator of bacterial metabolic activity. The use of TMA as a biomarker precludes possible contamination due to positive detection because of its origin. UPEC can also use l-serine dehydratase to break down urinary l-serine into ammonium and pyruvate and then use the synthesized breakdown products to satisfy carbon and nitrogen requirements [88]. In the meantime, it was discovered that UPEC obtains amino acids and that, in order to adapt in vivo, it needs glycolysis and the tricarboxylic acid cycle (TCA). The TCA cycle, peptide import, gluconeogenesis, the Entner–Doudoroff pathway, the oxidative and nonoxidative branches of the pentose phosphate route [89], and peptide import are all required for *E. coli* adaptation during glycolysis during UTI [90]. Short peptides are taken up by UPEC and converted into amino acids in order to attain maximal growth during infection. After that, these amino acids undergo catabolization and are used in a series of procedures that produce glycolytic substrates and renew intermediates in the TCA cycle [90]. Therefore, one of the main factors contributing to the harmful microbial pathogenicity of bacteria is their core metabolic pathway.

Iron carrier biosynthesis has been the subject of several investigations, since it is believed to be crucial to UPEC’s capacity to induce UTIs [91,92,93]. In contrast to nonpathogenic *E. coli*, iron transporters are often lacking in pathogenic *E. coli* strains [54]. Iron carrier biosynthesis was confirmed by metabolomics to synergistically regulate toxicity-related interacting metabolomes and could significantly differentiate between UPEC and non-UPEC [93]. Yan et al. used a targeted metabolomics method with a genetic strategy to learn how the high-pathogenicity island (HPI) regulates central carbon metabolism in UPEC. They found that, in comparison to non-UPEC K12 strains lacking the HPI, UPEC with the HPI exhibited superior metabolic homeostasis, which made it easier for UPEC with the HPI to adapt to a variety of growth settings [94].

Some bacteria produce specific metabolites: *Pseudomonas aeruginosa* (*P. aeruginosa*) metabolizes nicotinic acid to 6-hydroxynicotinic acid (6-OHNA) [95]. *Klebsiella pneumoniae* (*K. pneumoniae*) utilizes glycerol as the sole carbon source and electron acceptor under anaerobic conditions and produces 1,3-propanediol (1,3-PD) [96]. It is feasible to characterize these bacteria by their selective metabolism of glycerol. By measuring 6-OHNA and 1,3-propanediol using ^1^H NMR spectroscopy, it is possible to objectively determine related UTI. In urine samples, *E. coli* breaks down lactose into lactate, succinate, acetate, and ethanol. It was discovered that *E. coli* produces lactic acid as a distinct metabolite; other Gram-negative bacteria that cause UTIs only create succinic acid, acetate, and ethanol [97]. Other bacteria that cause UTIs do not metabolize methionine in urine; *Proteus mirabilis* (*P. mirabilis*) exclusively metabolizes methionine to 4-methylthio-2-oxobutyric acid (MOBA) [97] (see Figure 2, four bacteria-specific metabolic processes). In comparison to traditional approaches, the identification of metabolites, particularly by metabolomics, has extremely excellent sensitivity and false-negatives according to data from in vitro and ex vivo experiments (on patient urine samples).

## 4. Interactions and Regulatory Relationships between the Microbiome and Metabolome during UTI

### 4.1. Effects of Metabolites on Microorganisms

Human urine, with many bands that resemble chemical molecules, is a complicated fluid. Microorganisms are found in certain ecosystems, but they also need certain metabolic processes and physiological adaptations to live and thrive there [98]. However, most sources of high-energy nutrients are already depleted by host metabolism. Notably, bacteria entering the UT must first become acclimated to grow in the urine in order to cause UTI, and one of the most significant indications of pathogenic potential within the UT is the combination of flexible metabolism and fast growth in urine [99]. Carbohydrates are the preferred carbon source for many microbes, although urine typically lacks them. As a result, bacteria with a strong ability to exploit a variety of nucleic acids, amino acids, and secondary catabolic metabolites as sustenance have a distinct advantage over other species.

The current body of research [100] indicates that UPEC provides energy for other downstream processes, such as gluconeogenesis and the TCA cycle, by mostly using the catabolism of amino acids as opposed to carbs. Relying on acid-tolerant mechanisms, the deamination of l-serine to pyruvate and ammonia promotes the survival of *E. coli* under acidic conditions. The focus of research is now turning to the metabolic pathways required for microbial communities to function in ecological niches [101]. Variations in the UT microbiota’s composition are linked to changes in the microbiological environment of the UT, such as bladder urine oxygen tension [102]. Additionally, the UT needs a replenishable source of nutrients to maintain microbial development [82,103]. Fresh urine input provides nutrition for resident bacteria [104]. Moreover, almost half of these amino acids may be used as a nitrogen source and the other half as a carbon source by a variety of human commensal bacteria, such as *E. coli*, which are known to colonize the UT [82,105]. Enzymes expressed by *Lactobacillus*, *Bifidobacterium*, and *Streptococcus* degrade a variety of extracellular glucose polyamines into smaller metabolizable sugars [106,107,108]. Therefore, studying the metabolic pathways and their products that promote rapid urine growth may provide new insights into the fight against UTIs.

### 4.2. Regulation of Metabolites by Microorganisms

Mutations in urine growth induction from glycolytic metabolism to amino acid intake and metabolism are often observed in genome-wide screens for UT infections, indicating that the capacity of amino acids to be used as a source of carbon and nitrogen is critical for UT colonization. Asparagine, lysine, glycine, glutamine, histidine, alanine, threonine, and serine are often the most prevalent amino acids in urine [109]. Due to its need to modify its metabolism in order to live in such disparate microenvironments, UPEC is subjected to a variety of metabolic selection processes in the UT [110]. Within intracellular bacterial communities (IBCs), UPEC specifically catabolizes sorbitol and β-galactoside, serving as an illustration of this distinct metabolic control [111,112]. In order to control osmotic pressure and preserve cell volume, host bladder epithelial cells import and concentrate sorbitol [113]. In order to create glucose, UPEC can metabolize the host cell’s intracellular sorbitol supply [13]. Purines are produced by UPEC via salvage or de novo biosynthetic pathways in the meantime [100]. Similarly, intracellular β-galactose may be taken up and cleaved by UPEC to create galactose, which can then be metabolized via the DeLey–Doudoroff pathway or transformed to glucose [113,114]. Furthermore, in order to maintain the functioning level of acetyl phosphate, which is implicated in several two-component regulatory networks controlling the stress response and production of virulence factors [115], regulation of acetate metabolism is also crucial in UPEC [116]. Based on the availability of acetate and acetate-producing substances in the environment, UPEC can alternate between acetate synthesis and consumption. It largely controls acetate flow through the phosphate acetyl-transferase-acetate kinase pathway [117]. Bacterial strain colonization and the risk of UTI and rUTI are influenced by the gut–bladder axis. The presence of gut microbial taxa that are anti-inflammatory and that are linked to rUTI susceptibility may be mediated by butyrate-producing bacteria and short-chain fatty acid (SCFA)-producing bacteria. Survival and persistence in urine seem to depend on the production of threonine and the metabolism of long-chain fatty acid (LCFA) [118].

### 4.3. The Significance of Microbiome and Metabolome Synergy in the Development and Recovery of UTIs

UTI formation is facilitated by the interaction of the metabolome and microbiota [119]. Firstly, changes in the microbiome can lead to UTIs [22,120]. Normally, the microbiome in the urethra and bladder is relatively stable and helps to resist invasion by pathogenic microorganisms [121]. However, changes in the microbiome, such as antibiotic use [122] and decreased immune function [123], can lead to overgrowth of harmful microorganisms [124], thereby triggering UTIs.

Secondly, changes in the microbiome and metabolome can influence the severity of UTIs and the recovery process [22,100]. For instance, l-ornithine secretion by *E. faecalis* increases arginine production in *P. mirabilis*, a metabolic interaction that significantly reduces infection severity [125]. Microbial biofilms are critical in the origin and pathophysiology of catheter-associated UTIs. It has been discovered that patients with UTIs have different microbiomes and metabolomes from those of healthy people. For example, bacteria such as *E.coli* and *Staphylococcus aureus* [83,126] were commonly found in the microbiome of patients with UTIs, whereas *Lactobacillus* predominated in the microbiome of healthy individuals [22]. Bacterial metabolism contributes to their ability to cause infections [114], and common metabolites in the metabolome of patients with UTI include amino acids [88,114,127], acetyl phosphate [115], and polyamine [85], whereas normal metabolites predominate in the metabolome of healthy individuals. These microbiome and metabolome changes can reflect the extent of UTI and the intensity of the inflammatory response and are important for guiding treatment and determining prognosis.

Lastly, one treatment option for UTIs may include changing the microbiota and metabolome. The symptoms of UTIs can be alleviated and recurrences can be avoided by controlling the balance of the microbiome, for example, by utilizing probiotics, antimicrobial peptides, bacteriocins, and phage [128]. This will also increase the number of helpful bacteria and restrict the growth of dangerous microorganisms. At the same time, regulating the balance of the metabolome, e.g., by the use of anti-inflammatory drugs and by improving nutritional status, can reduce the inflammatory response and promote the repair and recovery of UT tissues.

## 5. Summary and Outlook

### 5.1. Potential Applications of the Metabolome in the Diagnosis and Prediction of UTI

Infections are caused by a variety of microorganisms that consume nutrients and secrete metabolic waste. As a result, these microbial metabolites could be prevalent near the infection site [129], making them potential targets for diagnostic techniques based on metabolism. Because bacteria metabolites are concentrated in the bladder and are readily detectable in the patient’s urine using methods like LC-MS and ^1^H NMR, this metabolic diagnostic approach is especially well-suited for UTIs [74,130,131]. Diamines, polyamines, and acylated couplings of these molecules are produced by *E. coli*, the most frequent pathogen associated with UTIs. These molecules are not often seen in human urine. Thus, these compounds have been suggested as possible UTI indicators [132]. Gregson et al. [133] used an untargeted metabolomics screen to identify two metabolites, guanidine-butylamine and N6-methyladenine, which were predictive of a variety of pathogen species that together accounted for almost 90% of UTIs. Fascinatingly, these metabolites have been reliably reported in several different cohorts. Therefore, by using metabolomics, we can quickly screen for the emergence of UTIs and enhance the treatment of antibiotics based on the presence of catabolic metabolites in urine.

### 5.2. Potential Applications of the Microbiome in the Diagnosis and Prediction of UTI

Treatment options for UTI may need to take the urine microbiota’s health into account as our understanding of its function in genitourinary health improves [22]. The conventional objective of obtaining UT sterility in the therapy of UTIs may disturb both infections and good, protective microbial communities. Without a healthy microbiota, the UT might be left in a sensitive, dysbiotic state where uropathogens could colonize it. The urethra’s *Actinomycetes* and *Mycobacteria* may provide a protective function by preventing the expansion of harmful organisms, and the quantity of organisms found throughout the body might function as a biological barrier [134]. Our study found that the number of bacterial species present in the urethra of female patients with UTI was reduced, with UPEC being present the most, followed by *Klebsiella pneumoniae*, *Staphylococcus saprophyticus*, *Enterococcus faecalis*, etc. However, the urethra of healthy females was rich in bacterial species, with the highest number of *Streptococcus*, followed by *Bifidobactenium*, *Staphylococcus*, *Micrococcus*, etc. Specifically, *Lactobacillus* in the female genitourinary tract is thought to be crucial in preventing UTIs because it can disrupt the adherence, proliferation, and colonization of uropathogenic bacteria. The overuse of antibiotics will change the composition of the bacterial flora, damage the biobarrier, and increase the amount of dangerous bacteria that are extremely resistant to drugs. In conclusion, a more comprehensive test to monitor the distribution and variability of the urogenital microbiota could help to better monitor the activity of pathogenic organisms and, hopefully, lead to early prevention. In addition to the composition of the microbiota, urogenital microorganisms can provide a wealth of information on antimicrobial susceptibility, which is an important guide for anti-infective treatment.

### 5.3. Application and Advantages of Emerging Therapies in UTI Research

#### 5.3.1. Mannoside FimH Antagonist

Most uropathogens cause infection through adhesion, colonization, and settlement [135], e.g., UPEC uses type I bacterial hairs to mediate binding to mannosylation receptors to colonize the bladder and invade uroepithelial cells in the lining of the bladder lumen [136]. However, the presence of tiny sugar molecules that compete with the receptor for contact by filling the binding pocket of bacterial hair adhesins can prevent bacterial attachment to sugar groups on glycoproteins [137,138]. After the structure–activity link was clarified clinically, mannosides—oral biologics that block FimH function—were developed. Within just 6 h of treatment, these mannosides have been shown to decrease bladder bacteria by four log units in a rat model for UTI [139]. Unlike antibiotics, which reduced the complexity of the gut microbiota, mannosides did not alter the native microbiota’s structure in mice models for acute cystitis and gastrointestinal flora [137].

#### 5.3.2. Probiotic Therapy

According to recent research, probiotics may be helpful for women who have a history of complex UTIs or long-term antibiotic usage. Furthermore, probiotics prevent antibiotic resistance [140]. Studies have shown a strong correlation between the microbiota in urine and the gut, indicating that controlling the gut flora may be a viable tactic for treating or preventing UTIs. Restoring a healthy urogenital microbiota may play a significant role in the diagnosis and treatment of UTIs in the future [37]. Following up on patients who had no extended-spectrum β-lactamases (ESBLs) and *K. pneumoniae* in their urine and feces and recurrent *Clostridioides difficile* infections, Grosen et al. [141] reported that fecal microbiota transfer (FMT) was a successful treatment intervention. Probiotics have additional benefits that also apply to the vaginal microbiome. Research has demonstrated that vaginal estrogen cream and probiotics, including *Lactobacillus* species, can be used together to treat UTIs [142]. Another study found that oral probiotics plus low-dose estrogen significantly decreased the incidence of rUTI in postmenopausal women [143]. *L. rhamnosus*, *L. reuteri*, *L. acidophilus*, and *L. casei* are among the other *Lactobacill* strains that are thought to be viable candidates for therapeutic use [128,144].

#### 5.3.3. Transfer of Fecal Microbiota

Research has demonstrated that FMT—which involves transferring microbes and metabolites from healthy donors to patients—can effectively disrupt the recurrence pattern, with response rates as high as 80% following a single dosage when administered via oral capsules, rectal enemas, or endoscopy [145]. A registry of patients who underwent FMT in order to show signs of recurrent *C. difficile* infection provided the first data on the effect of FMT on UTI. Prior to FMT, a sizable portion of these patients had recurring *C. difficile* infections together with UTIs. Among the individuals who received FMT, the incidence of UTIs was markedly decreased [59]. By producing antimicrobial compounds, competing for resources, adjusting intestinal pH, and preventing access to binding sites in the mucosal wall, symbiotic enteric bacteria contribute significantly to resistance against pathogen colonization [146]. Though the exact mechanism of action is yet unknown, it is quite likely that UT pathogens that have colonized feces reservoirs would be eliminated by the transfer of fecal microbiota, thereby removing reinfection sources.

### 5.4. Future Research Needs and Possible Challenges

By comprehending the flora and metabolites, we have discovered that using antibiotics to change the microbiota without any thought might raise the risk of infection and that maintaining a healthy urogenital microbiota could play a significant role in UTI treatment in the future. In the study of monitoring the urogenital flora, from pathogenesis to treatment, it is no longer limited to a single causative organism, and clinicians in the diagnosis and treatment of UTIs should not be confined to the use of antibiotics to fight against the causative organisms but should elevate the diagnostic and treatment focus to the level of the rich bacterial flora to break through the bottlenecks encountered in the diagnosis and treatment of UTIs; it is expected that important progress will be made in the rational use of antibiotics.

Thanks to NGS technologies, professionals may now more accurately identify the microorganisms causing UTIs. Professionals are also able to monitor changes in the resident flora, which helps them better understand the onset, progression, and prognosis of UTIs. This understanding is gaining ground as more attention is paid to the combined effects of multiple organisms and the potential impact of changes in the original flora. Macrosequencing may be a powerful aid in exploring new frontiers. However, macroscopic sequencing also suffers from the problem of results that are “too comprehensive”. For example, using NGS, it is difficult to determine which of the many bacteria ultimately detected in urine are present in the urine or contaminated during sampling [147]. The synergistic work of the metabolome and microbiome may revolutionize the clinical management of UTIs.

## Figures and Tables

**Figure 1 ijms-25-03134-f001:**
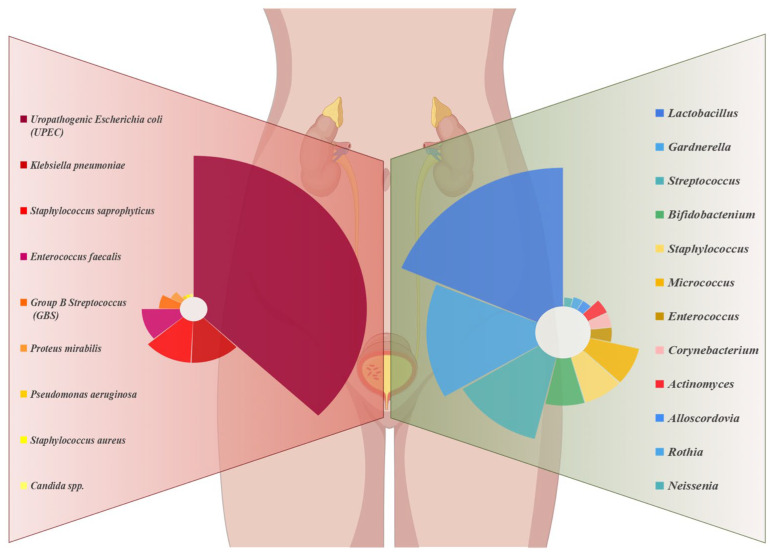
Summarizes the urethral microbiota of healthy female individuals (**right**), and of UTI patients (**left**).

**Figure 2 ijms-25-03134-f002:**
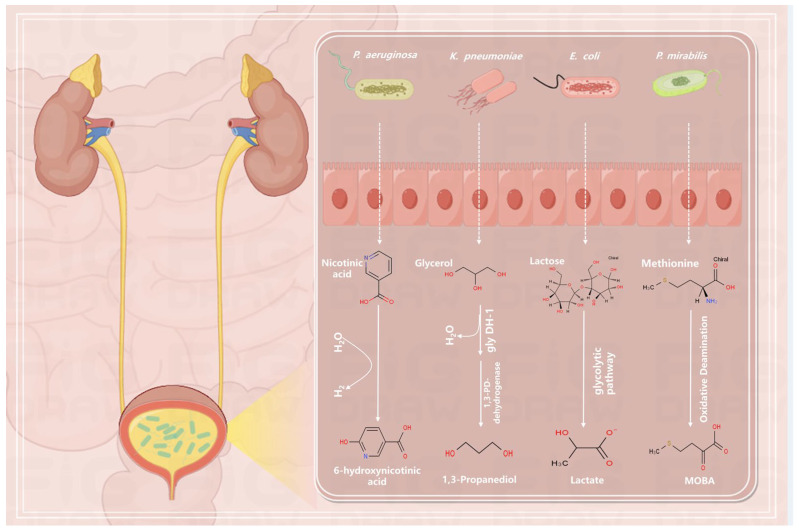
Four bacteria-specific metabolic processes; *P. aeruginosa* metabolizes nicotinic acid to 6-OHNA by de-H. Glycerol is first produced as 3-hidroxypropanaldehyde (3-HPA) by glycerol dehydratase, which is dehydrogenated to dihydroxyacetone (DHA) through the oxidative pathway and then phosphorylated to dihydroxyacetone phosphate (DHAP) by ATP-dependent kinases. By a parallel reduction pathway, glycerol is dehydrated by a B12-dependent enzyme to form 3-hydroxypropionaldehyde, which is then reduced to 1,3-PD by an NADH-linked oxidoreductase (1,3-PD dehydrogenase). Lactose is catabolized and metabolized to galactose and glucose by β-galactosidase, and, in the Embden–Meyerhof–Parnas (EMP) pathway, glucose (galactose) is reduced to lactate by lactate dehydrogenase after the production of pyruvate when the cell is unable to obtain enough oxygen or process it fast enough; *P. mirabilis* undergoes oxidative deamination to metabolize methionine to MOBA.

**Table 1 ijms-25-03134-t001:** Pathogenesis of catheter-associated UTIs.

Species	Main Virulence Factors	Function
*Enterococcus*	Ebp pili	Fibrinogen-binding adhesin of EbpA binds to fibrinogen of encapsulated catheters
	SprE	Secreted proteases (GelE and SprE) to cleave fibrinogen to enhance biofilm formation
	GelE	
*Staphylococcus*	ClfB	Binding fibrinogen, thereby promoting bladder and duct colonization
*Proteus*	MR/P fimbriae	Adhesion to the bladder establishes infection
	Urease	Hydrolysis of urea to ammonia, formation of crystalline biofilms, provision of protective ecological niches and promotion of relapse
*Escherichia*	FimH	FimH adhesin binds fibrinogen covering the duct

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
