# Peer review of "The Role of Metabolomics and Microbiology in Urinary Tract Infection"

_ijms, 2024, doi:10.3390/ijms25063134_

Round 1
Reviewer 1 Report
Comments and Suggestions for Authors
The authors presented a Review document entitled "The Role of Metabolomics and Microbiology in Urinary Tract Infection", the hace recovered a huge amount of interesting data on the field of diagnostic microbiology.
I Have some suggestions:
L78. "et al" must be with period at the end "et al" -> "et al.", please revise the entire document.
L93. Citation must be before the period "age. [26]" -> "age [26]."
L120, L174-L176, L186-187, 233-234, 240-241. Please rephrase the sentence to make it clear, in general the entire document must be verified by a native English speaker.
L125 and L129. Please change the parenthesis sign before de period.
L150, L371, etc. All the abbreviations used in the document must be clarified when are used for the first time (rUTI, IBC, SCF, LCF).
L174. Erase the period before Enterococcus.
L176-177, 357 please erase the period after de "sp" or "spp", those words should not be with period unless are at the end of a sentence.
Table 1. Please turn the table into an adequate format trying to fit the species name in the cell, please shift the bacteria names in italicized shape. Pleas change "Escherlchla" for "Escherichia".
L247. Nuclear Magnetic Resolution? or Nuclear Magnetic Resonance?
L281. When you are mentioning genes names, it must be italicized.
L427, 273, 300, 311, 348, etc. Please verify the entire document, there are several issues regarding the bacteria names, they must be italicized.
L312. Please shift Pr. mirabilis to P. mirabilis
L320. Please shit "3hydroxypropanaldehyde" to "3-hidroxypropanaldehyde"
Comments on the Quality of English LanguageThe document require an exhaustive revision by a a native English speaker
Reviewer 2 Report
Comments and Suggestions for Authors
I have read with interest the manuscript submitted by the authors since this topic represents a great concern worldwide.
I have a few comments to be addressed in order to improve the quality of the manuscript:
preventive treatment - if we talk about treatment, it is not prevention
many phrases are too long, making them hard to understand.
once described an abbreviation, such as UTI - only the abbreviated form should be used
row 62 - this information is not new at all..
row 85 - Gardnerella follows closely in terms of abundance. However, fig 1 states differently.
Fig 1 - all abbreviated terms should be described.
row 108 - false-positive diagnosis - maybe false-negative
many abbreviations are not described at all, such as: WGS, UPEC, rUTI, TCA, IBCs and many more
Moustafa et al [30], 116 samples were successfully analysed by 16S rDNA sequencing and grouped according to clinical laboratory clusters - please rephrase
Bacterial names are miswritten all over the manuscript;
Fig 2, even though visually looks good, has a low resolution and the text is very hard to understand
rows 355-357 - please rephrase
it is not indicated to start a phrase with the word "and"
The conclusions of the study should be clearly presented.
Overall, the study seems to be disorganized, containing a great amount of information but not presented clearly.
Comments on the Quality of English Languagemany misspells, and punctuation misuse.
Round 2
Reviewer 1 Report
Comments and Suggestions for Authors
All suggestions have been properly addressed and the manuscript has been considerably improved.
Reviewer 2 Report
Comments and Suggestions for Authors
I appreciate the author's efforts in addressing my comments. The quality of the manuscript has increased. Only some minor changes are still required:
Moustafa et al. [30], 116 samples were successfully analysed by 16S rDNA sequencing, found that abnormal proliferation of microbiota such as Aspergillus and E. coli was found in the patients - Moustafa et al. [30] successfully analyzed 116 samples by 16S rDNA sequencing and found that ....
Clostridium -> Clostridioides
Please add a single phrase/paragraph at the end with the most important findings of your research.
For future publications - I highly recommend the authors use the track changes feature to show the exact manner in which the manuscript has been revised.
Best regards,
Comments on the Quality of English Languageminor
